# Impact of Tumor Size on Prolactinoma Characteristics and Treatment Outcomes: A Study of a Tunisian Cohort

**DOI:** 10.3390/biomedicines13051125

**Published:** 2025-05-06

**Authors:** Mouna Elleuch, Hamdi Frikha, Fatma Loukil, Khouloud Boujelben, Dhouha Ben Salah, Nabila Mejdoub Rekik

**Affiliations:** Department of Endocrinology, Hedi Chaker University Hospital, Faculty of Medicine of Sfax, University of Sfax, Sfax 3029, Tunisia; frikha.hmd@gmail.com (H.F.); fatma.loukil@aphp.fr (F.L.); khouloud25boujelben@gmail.com (K.B.); bs.dhoha@gmail.com (D.B.S.); nabila.rekik.mejdoub@gmail.com (N.M.R.)

**Keywords:** prolactinoma, tumor size, prognosis

## Abstract

**Issues:** The clinical and paraclinical characteristics of prolactinomas differ mainly according to sex and tumor size. Drug treatment with dopamine agonists (ADs) has a crucial role in the management of prolactinomas. The use of surgery also has its indications. **Purpose of the work:** We aimed to establish the therapeutic strategy and the follow-up profiles of prolactinoma while analyzing the predictive factors of remission; we also looked for correlations between the size of the prolactinoma and the various clinical and paraclinical parameters. **Materials and methods:** This was a retrospective, descriptive, and analytical study of 77 cases of prolactinomas collected and monitored at the endocrinology and diabetology department of the Hedi Chaker CHU in Sfax between 2000 and 2017. Our patients were divided into three groups according to the size of their prolactinomas. Statistical correlations were sought between tumor size and clinical and biological parameters. **Results:** The mean age of our patients was 38.3 ± 14.2 years. The sample comprised 51 women (66.2%) and 26 men (33.7%). Anterior pituitary syndrome was observed in 75.3% of cases. The number of antehypophyseal deficits was significantly correlated with tumor size. Comparing the three groups, we noted that age, discovery circumstances, metabolic parameters, hypopituitarism, and pituitary extensions on imaging were significantly different. Therapeutically, our results showed that the favorable evolution of prolactinomas was correlated with tumor size and the duration of treatment. **Conclusions:** Tumor size appears to be a cornerstone in hormonal and radiological interpretation on the one hand and in the therapeutic decision on the other.

## 1. Introduction

Prolactinoma is the main etiology of pathologic hyperprolactinemia [1]. Depending on the size of the tumor, we can distinguish microprolactinomas which are the most common (90% of cases). Macroprolactinomas are rarer. Among the latter, a distinction is made for giant prolactinomas, which are larger than 40 mm [2,3]. Macroprolactinomas are seen more frequently in men. Drug treatment with dopamine agonists (ADs) has a crucial role in the management of prolactinomas. However, the therapeutic strategy for prolactinomas has hitherto remained mainly dependent on the size of the tumor. The prognosis of these patients is not well established since it depends on several factors. In this context, we conducted this study to establish the prognostic factors of prolactin adenomas. The main objective of the study was to look for correlations between the size of the prolactinoma and the various clinical, paraclinical, and follow-up parameters.

## 2. Methods

This study is a retrospective, descriptive, and analytical investigation conducted at the Endocrinology Department of Hedi Chaker University Hospital in Sfax. It focused on cases of prolactinomas collected and monitored over an 18-year period, from 1 January 2000 to 31 December 2017.

### 2.1. Study Design and Population

We included all patients diagnosed with prolactinomas during the study period. Diagnosis was established based on elevated prolactin levels and confirmatory imaging findings on magnetic resonance imaging (MRI). To ensure consistency, patients with incomplete medical records or insufficient follow-up data were excluded.

The study population was divided into three groups based on tumor size, as determined by MRI:

**Group 1 (G1):** Microprolactinomas (<10 mm in size)—27 cases (35%).

**Group 2 (G2):** Macroprolactinomas (10–40 mm)—32 cases (41.6%).

**Group 3 (G3):** Giant prolactinomas (>40 mm)—18 cases (23.4%).

### 2.2. Main Steps of the Study

The main steps of our study consisted of the following:A descriptive overview of the clinical, hormonal, and radiological features of prolactinomas and their comparison between the three groups.An evaluation of treatment strategies and outcomes across the three groups.An analytical study to identify predictive factors of prolactinoma remission.

### 2.3. Data Collection

For each patient, detailed clinical, biological, imaging, and anthropometric data were collected.

#### 2.3.1. Clinical Parameters

Data on the presenting symptoms (e.g., galactorrhea, amenorrhea, infertility, or visual disturbances), demographics (age and sex), **and anthropometric parameters** including body mass index (BMI), weight, and height were collected.

#### 2.3.2. Biological Parameters

The hormonal profile included serum prolactin, follicle-stimulating hormone (FSH), luteinizing hormone (LH), testosterone, estradiol, and cortisol levels.

Central hypothyroidism is defined as low free T4 with inappropriately low or normal TSH.

Hypogonadotropic hypogonadism is defined as low sex hormone levels (testosterone in men; estradiol in women) with low or inappropriately normal gonadotropin levels (FSH and LH).

Corticotroph deficiency is defined as low morning cortisol levels (<5 µg/dL) or insufficient response to a standard ACTH stimulation test.

A metabolic panel includes total cholesterol, high-density lipoprotein (HDL) cholesterol, triglycerides, and fasting glucose.

#### 2.3.3. Imaging Parameters

Imaging parameters included tumor size, location, and invasion based on MRI findings.

### 2.4. Treatment and Evaluation of Outcomes

We evaluated the indications and outcomes of both medical and surgical treatments. Medical treatment primarily consisted of dopamine agonists (DAs) such as bromocriptine, cabergoline, and quinagolide. Surgical intervention was considered for patients with DA resistance, intolerance, or complications requiring immediate tumor debulking.

### 2.5. Definitions of Treatment Outcomes

**Remission:** Defined as the disappearance of the prolactinoma on MRI with a normal prolactin level sustained for at least 24 months with minimal dose DA therapy.

**Favorable Outcome:** Characterized by the normalization of prolactin levels and tumor size reduction by more than 50% upon a follow-up MRI.

**Relapse:** Categorized as early (re-increase in prolactin >30 ng/mL within one year of normalization) or late (re-increase >30 ng/mL after one year of normalization).

**Resistance to Dopamine Agonists (DAs):** Defined as failure to achieve prolactin normalization or tumor size reduction ≥ 50% after three months of DA therapy at maximum tolerated doses (15 mg/day bromocriptine, 4 mg/week cabergoline, or 300 mg/day quinagolide).

**Dosage Timing:** Hormonal and imaging evaluations were conducted at regular intervals: 1 month, 6 months, 1 year, 5 years, and 10 years post-initiation of treatment. This enabled tracking of both short-term and long-term outcomes.

### 2.6. Statistical Analysis

The data entry was carried out on a computerized form by the SPSS software (version 20). A correlation study was carried out in this work between the different qualitative and quantitative variables. The difference between the results was considered significant when the *p*-value was <0.05. The link between two continuous variables was tested by Pearson’s correlation in the case of Gaussian distribution and by Spearman’s correlation whenever the normality of the distributions was not respected. We also used the Chi-square test for the correlation between the qualitative values with recourse to the Fisher test in the event of small numbers. Student’s tests and ANOVA tests were used to seek a correlation between quantitative variables.

## 3. Results

### 3.1. Epidemiological Data

Our study involved 77 cases of prolactin adenomas including 26 men (33.7%) and 51 women (66.2%). The mean age of our patients was 38.3 ± 14.2 years with extremes ranging from 14 to 75 years. Our results showed that the larger the tumor, the more advanced the age of diagnosis. Indeed, in G1, the mean age was 31.4 years versus 41.5 and 42.9 years in G2 and G3, respectively. The statistical study showed the existence of a positive and significant correlation between the age of discovery and the size of the prolactinoma (*p* = 0.003 and Pearson’s r coefficient = 0.348).

### 3.2. Anthropometric Data

Obesity was more common in G3. Paralleling this finding, mean waist size was significantly higher in G3 irrespective of sex. Anthropometric data are summarized in Table 1.

### 3.3. Biological Data

#### 3.3.1. Metabolic Parameters

Metabolic syndrome and carbohydrate tolerance disorders were significantly more frequent in G3 versus G1 and G2. (Table 2).

#### 3.3.2. Prolactinemia

The mean prolactinemia was higher in G3 (10,569.1 ng/mL) versus G2 (523.4 ng/mL) and G3 (164.1 ng/mL) with a statistically significant difference (*p* = 0.002). This same result was also demonstrated with the medians (Figure 1).

Overall, the tumor size category impacted prolactinemia level distribution (*p* = 0.002). Prolactinemia < 100 ng/mL was more frequently reported in G1 (78.3%) with a statistically significant difference (*p* < 0.005). Only three patients (11.5%) of G1 presented prolactinemia > 200 ng/mL. Concerning prolactinemia between 100 and 200 ng/mL, the values were equally distributed between the three groups. Prolactinemia of between 200 and 1000 ng/mL was observed mainly in G2 (84.2%) (Figure 2). Prolactinemia > 1000 ng/mL was significantly more common in group 3 (*p* < 0.005).

An overall statistically significant positive correlation was found between the size of the prolactinoma and the prolactinemia (*p* = 0.001 and r = 0.4). Intriguingly, this correlation was not significant when tested in each group separately (Figure 3).

#### 3.3.3. Other Hormonal Dosages

In both sexes, the mean FSH and LH levels were lower in G3 versus G1 and G2 (*p* < 0.05). Mean testosteronemia in men was also lower in G3 versus G1 and G2 but this difference was not significant (Table 3).

The comparison between the three groups showed that the mean number of antehypophyseal deficits was significantly greater in G3 (2.17) versus G2 (1.25) and G1 (0.70) with a statistically significant difference (*p* = 0.008). The number of prolactinomas with only one affected axis was higher in G1 (14 cases) versus G2 (12 cases) and G3 (3 cases). A statistically significant difference was found concerning the frequency of each anterior pituitary deficit in the three groups (Table 4).

In each group, it was observed that the more the size of the prolactinoma increased, the more the number of AH deficits was important (*p* < 0.001). This finding was irrespective of the group.

### 3.4. Treatment

Recourse to medical treatment alone was more frequent in G1 (92.1%) versus G2 and G3 (62.5% and 61.1%, respectively). A positive and statistically significant correlation was established between the duration of treatment and the size of the prolactinoma (*p* = 0.020). The use of surgical treatment was greater in G2 and G3 (37.5% and 38.8%) versus G1 (7.4%). The difference was statistically significant between the three groups (*p* = 0.024) (Table 5).

### 3.5. Follow-Up Data

Normalization of prolactinemia was observed in 47 patients (61%) after an average of 16.5 months (1–120 months). It mainly depended on tumor size and duration of treatment. In fact, it was more frequent in G1 (70.3%) versus G2 (59.3%) and G3 (38.8%) patients. At one month of treatment, there was no difference in the normalization of prolactinemia between the groups. Beyond 6 months, G1 and G2 patients tended to normalize their prolactinemia more frequently than G3. This difference reaches statistical significance after 5 years. (Table 6).

Early and late relapses were observed in 9 (11.4%) and 15 patients (19.4%), respectively. Remission was observed in 10 patients (13%). It was more common in patients treated with CB (50%), those who were female (80%), and those with microprolactinomas (40%) (Table 7). Resistance was found in 10 patients (13%). It was more frequent in patients treated with BC (100%), those who were female (60%), and those with macroprolactinomas (70%) and giant prolactinomas (30%) (Table 7).

Regarding the predictive factors of remission, the reduction in tumor size by more than 50% depended, on the one hand, on the initial tumor size (*p* = 0.001) and, on the other hand, on the duration of treatment (*p* = 0.000) (Table 8).

## 4. Discussion

A statistically significant positive correlation was revealed between the age of discovery and the size of the prolactinoma. This is also confirmed by a large study carried out in Iceland [4,5,6,7] where the age difference between macroprolactinomas and microprolactinomas was 10 years (42 years versus 32.5 years, respectively).

In our study, the frequency of the male sex gradually increased with the size of the adenoma; it increased from 11.2% in G1 to 77.7% in G3. This is explained in the literature by the fact that the adenoma has higher proliferative indexes (ki67 and PANCA) in men than in women [8,9,10,11]. Other studies also show that the activity and expression of transforming growth factor B1 (TGFB1) is elevated in men, leading to increased size and aggressiveness of the prolactinoma [8,12,13].

In our study, the mean initial prolactinemia was 2745.9 ng/mL (26–81,940 ng/mL). It was 933.5 ng/mL (32–21,200 ng/mL) in a Brazilian multicenter study including 250 macroprolactinomas and 444 microprolactinomas [14]. There is a close relationship between the secretory level and the size of the adenoma [11,14,15]. In our population, a statistically significant positive correlation was observed between the size of the prolactinoma and the prolactinemia (*p* < 0.001). This result was also confirmed in the work of Chambeh et al. with 37 prolactinomas included in their study (*p* < 0.001) [16]. A positive correlation was also found between prolactinemia and tumor size within the three groups. This correlation is well established in the literature for giant prolactinomas (*p* < 0.001) [17,18,19,20].

The prognosis of prolactin adenoma also depends on their hormonal and metabolic impact. In this context, a statistically significant difference was found between the three groups concerning the frequency of each anterior pituitary deficit which was more frequent in G3. Another positive and significant correlation was demonstrated between the size of the prolactinoma and the number of anterior pituitary deficits that affected our patients (*p* < 0.01). Hormonal assays also demonstrated that the mean FSH and LH levels were significantly lower in G3 versus G1 and G2. Studies by Amit Tirosh and Sibal L [21,22,23,24] confirmed the same results.

On the metabolic level, in our study, obese and/or overweight patients were statistically more frequent in G3 versus G1 and G2. Likewise, waist size means were greater in G3 in both sexes. According to two studies in the literature [10,25], patients with a higher prolactin level have a higher BMI and WS. The correlation between these parameters is not significant. However, the decrease in BMI and WS after 2 to 6 months of treatment is significant in men [25,26,27]. In the literature, hyperprolactinemia is frequently associated with a carbohydrate disorder [10,25]. Several physiopathological phenomena have been suggested concerning this subject. Indeed, prolactin causes a state of insulin resistance (IR) and hyperinsulinism secondary to the increase in the cell mass of the islets of Langerhans [28]. Overall, metabolic syndrome (MS) was more frequently observed in G3 (58.8%) versus G1 (22.2%) and G2 (37.5%) patients with a statistically significant difference. The studies by Auriemma and Dos Santos Silva [25,27] confirm our findings.

The follow-up data of our patients showed that the success of surgery for microprolactinomas is estimated at 75% [15] and can reach 90% with trained surgeons [29,30,31,32]. For macroprolactinomas, the results of two prospective studies [33,34] show the normalization of prolactinemia under CB in 77% of cases and a significant reduction in tumor size in 92% of patients. The maximum effectiveness of CB is reported after 6 months of treatment. These results are not close to our findings since the majority of our patients were treated with first-line BC. The better efficacy of CB compared to BC is confirmed by statistically significant results according to Vroonen et al. [35].

Regarding giant prolactinomas in our study, medical treatment was weakly effective, unlike the studies by Maiter et al. [18] and Espinosa et al. [17] which showed that the giant tumor responds well to medical treatment. In fact, they noted a reduction of more than 25% in tumor size on average in 74% to 98% of cases and the normalization of prolactinemia in 55% to 60% of cases. This discrepancy with our results could be explained by the large number of patients lost to follow-up in G3. The disappearance of giant prolactinoma after medical treatment is described in the literature [36,37,38]. Indeed, some authors recommend a high dose of cabergoline (3.5 to 4.5 mg/week). Others offer long-term treatment for up to 20 years. In most of the studies [19,31,39,40], surgery for giant prolactinomas is generally associated with the persistence of hyperprolactinemia and a tumor residue requiring the use of ADs postoperatively. Early recurrence is reported in 45.9% of patients in the study of Sala E et al. [25]. This does not depend on tumor size or gender, but it is correlated (*p* = 0.03) with prolactinemia at the time of diagnosis and upon discontinuation of treatment. In a large meta-analysis including 19 studies [41], stable normoprolactinemia was demonstrated in 21% of microprolactinomas and 16% of macroprolactinomas after discontinuation of AD [4,5]. This assumes that the majority of patients suffer from a rebound effect of hyperprolactinemia upon discontinuation of medical treatment.

In the case of resistance to treatment, data in the literature suggest that this phenomenon is rather observed in men, macroprolactinomas, and patients treated with BC (25% versus 10% for CB) [22,35]. The mechanism of this resistance is explained by the decrease in the number of dopaminergic (D2) receptors in resistant prolactinomas [42,43,44]. A molecular alteration downstream of D2 or a difference in the expression of the isoforms of D2 (D2415/D2444) could also explain this phenomenon [43,45,46,47]. According to various studies [29,48,49,50], remission is significantly more frequent in microprolactinomas (between 65% and 78%) compared to macroprolactinomas (between 57% and 36%). According to Teixeira et al. [49], remission is influenced only by the initial tumor size. Other studies [48,51,52,53] confirm that female sex and the use of CB are considered to be factors promoting remission, which is consistent with our results. Concerning the predictive factors of remission, Hofstetter and Amar [54,55,56] demonstrated a statistically significant negative correlation between the rate of prolactinemia at day 1 postoperatively and the rate of remission. In a study by Catarina Araújo et al. [28], which included 67 patients, the normalization of prolactinemia and the reduction in tumor volume by more than 50% only depended on the total duration of treatment (*p* = 0.001) and the maximum dose of ADs (*p* = 0.019). Our results confirmed the role of the duration of treatment (*p* = 0.027) and the initial size of the adenoma (*p* = 0.001) in remission. This was not influenced by the maximum dose of ADs. In fact, the latter was not achieved in many of our patients mainly due to the lack of means.

## 5. Conclusions

Our study made it possible to highlight the metabolic and hormonal repercussions of prolactinomas. The latter were significantly associated with the occurrence of metabolic syndrome, abnormalities in glucose tolerance, and antehypophyseal deficits. These abnormalities were all the more important for larger tumor sizes. This work also made it possible to identify the predictive factors of remission of prolactinomas, namely the duration of treatment and a smaller initial size of the adenoma. These results should guide the initial assessment and therapeutic management of prolactin adenomas.

## Figures and Tables

**Figure 1 biomedicines-13-01125-f001:**
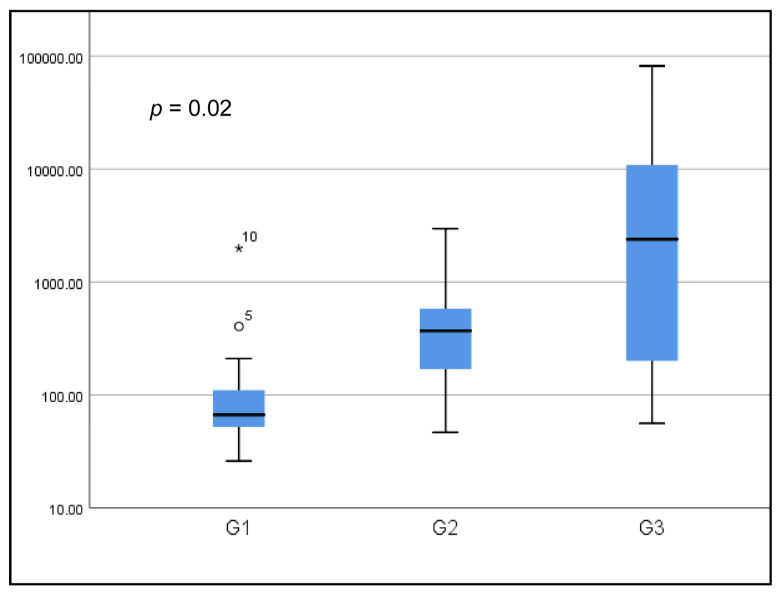
Boxplot of prolactin levels according to groups (logarithmic scale).

**Figure 2 biomedicines-13-01125-f002:**
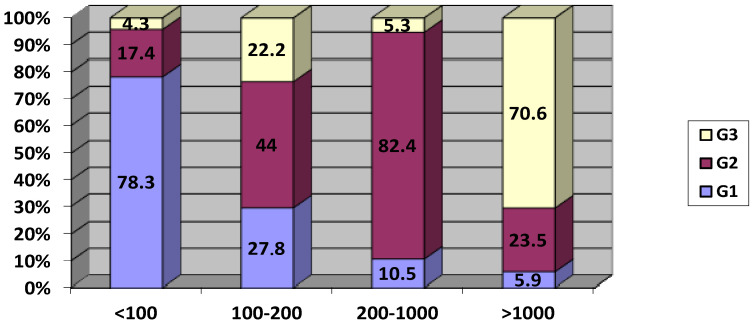
Distribution of tumor size category according to serum prolactin levels (ng/mL).

**Figure 3 biomedicines-13-01125-f003:**
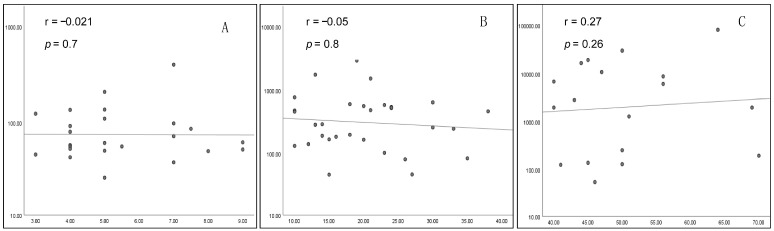
Correlation between tumor size (mm) and prolactinemia (log scale) ((**A**) G1; (**B**) G2; (**C**) G3).

**Table 1 biomedicines-13-01125-t001:** Anthropometric parameters by group.

	G1	G2	G3	Total	*p*
Overweight and/or obese	19 (70.3%)	19 (59.3%)	17 (94.4%)	55 (71.4%)	0.029
Mean waist size in women (cm)	87.3	95.6	113	93.8	0.006
Mean waist size in men (cm)	106	96.4	98.5	98.4	0.051

**Table 2 biomedicines-13-01125-t002:** Metabolic disorders by group.

	G1	G2	G3	Total	*p*
Dyslipidemia	14 (51.8%)	16 (50%)	12 (70.5%)	42 (54.5%)	0.454
FHG/ICH	0%	4 (12.5%)	5 (29.4%)	9 (11.6%)	0.042
Metabolic syndrome	6 (22.2%)	12 (37.5%)	10 (58.8%)	28 (36.3%)	0.042

FHG: moderate fasting hyperglycemia. ICH: intolerance to carbohydrates.

**Table 3 biomedicines-13-01125-t003:** Comparison of the means of FSH, LH, and testosterone between the 3 groups.

Mean	G1	G2	G3	*p*
FSH (IU/mL)	11.4	4.1	2.9	0.049
LH (IU/mL)	9.5	2.9	1.1	0.004
Testosterone (ng/mL)	1.4	1.9	0.9	0.266

**Table 4 biomedicines-13-01125-t004:** Frequency of anterior pituitary deficits by group.

	G1	G2	G3	*p*
Hypogonadotropic hypogonadism	14 (26%)	22 (42.3%)	17 (32.7%)	0.008
Central hypothyroidism	2 (14.3%)	5 (35.7%)	7 (50%)	0.028
Corticotroph deficiency	3 (9.4%)	14 (43.8%)	15 (46.9%)	0.000
Three-axis deficit	1 (9%)	4 (36.3%)	6 (54.5%)	0.002

**Table 5 biomedicines-13-01125-t005:** Frequency of treatment used by group.

	Medical Treatment Only	Surgical Treatment
G1	25 (92.5%)	2 (7.4%)
G2	20 (62.5%)	12 (37.5%)
G3	11 (61.1%)	7 (38.8%)
*p*	0.082	0.024

**Table 6 biomedicines-13-01125-t006:** Progression of prolactin levels by group.

	G1			G2			G3			
Dosage Timing	N	M	%	N	M	%	N	M	%	*p*
Pre-therapeutic	26	164.1	0	32	523.4	0	18	10,569.1	0	0.000
After 1 month	10	71.8	33.3%	11	689.7	33.3%	9	4594.9	33.3%	0.989
After 6 months	20	41.5	30%	23	97.8	50%	13	726.7	20%	0.598
After 1 year	18	56.1	50%	21	136	38.5%	5	1315	11.5%	0.283

N: number of assessed prolactinemia (PRL). M: mean PRL (ng/mL). %: percentage of normal PRL.

**Table 7 biomedicines-13-01125-t007:** Distribution of remission and resistance.

	Remission (N = 10)	Resistance (N = 10)
Men	4 (40%)	4 (40%)
Women	6 (60%)	6 (60%)
Under cabergoline	5 (50%)	0%
Under bromocriptine	0%	10 (100%)
Mean treatment duration before remission	6.2 ± 2.6 years	N/A
G1	4 (40%)	0%
G2	4 (40%)	7(70%)
G3	2 (20%)	3 (30%)

**Table 8 biomedicines-13-01125-t008:** Remission parameters as a function of tumor size, dose, and duration of treatment.

	Prolactin Normalization During Follow-Up		>50% Diameter Reduction	
	No = 32 (41.5%)	Yes = 45 (58.4%)	No = 42 (53.3%)	Yes = 35 (46.7%)
G1	8 (29.6%)	19 (70.3%)	17 (63%)	10 (37.3%)
G2	13 (40.6%)	19 (59.3%)	10 (31.2%)	22 (68.7%)
G3	11 (6.1%)	7 (38.8%)	15 (83.3%)	3 (16.6%)
*p*	0.076		0.001	
Mean maximal dose of BC (mg)	7.72	6.71	7.07	7.4
*p*	0.39		0.49	
Mean maximal dose of CB (mg)	0.91	0.68	0.78	0.79
*p*	0.16		0.780	

## Data Availability

The original contributions presented in this study are included in the article and Appendix A. Further inquiries can be directed to the corresponding author.

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
