# Peer review of "Impact of Tumor Size on Prolactinoma Characteristics and Treatment Outcomes: A Study of a Tunisian Cohort"

_biomedicines, 2025, doi:10.3390/biomedicines13051125_

Round 1
Reviewer 1 Report
Comments and Suggestions for Authors
The authors have performed a retrospective analysis of 77 cases of prolactinoma followed at their institution. They have divided the population into microadenomas, macroadenomas, and giant prolactinomas. As the analysis is presented it is very confusing, and as presented unsalvageable.
They might consider reevaluating the data along the lines of doing a regression/correlation analysis with the size of the tumor versus the level of the prolactin. They might consider the effect of the type of dopamine agonist drug with lowering of prolactin and decreasing the size of the adenoma. They need to limit the analysis up to one year. The analyses at 5 in 10 years does not consider dropouts. They may be able to perform a time course of prolactin over the year for G1, G2, G3.
The definitions need to be reconsidered. Remission would mean not requiring any medication after either medical or surgical treatment. It is not clear how many went into remission and did not require drug therapy, nor how long they were treated before they went into remission. Favorable outcome for tumor size, especially in larger tumors, must be evaluated at six to nine months, not at three months. Relapse would be someone who has been successfully treated with medication or surgery and has been stable and then has a recurrence. Resistance to dopamine agonists should be defined as resistance to achieving normal prolactin levels after three months. The effects of each dopaminergic drug should be evaluated. Resistance to tumor size reduction should be evaluated at six months and also according to the therapy. Teh time course to remission and relapse might b considered for G1, G2 ,G3.
Figure 1 should be replaced by a regression of tumor size versus prolactin levels showing mean and standard deviations.
Table 1 with the corresponding result section is meaningless without a hypothesis on weight. They have acknowledged that G3 is from an older group, and the observation without a hypothesis has no prognostic value. Table 4 also should be eliminated as there is no hypothesis for metabolic syndrome to separate the three groups.
Figures 3,4,and five are in French but also should be eliminated. and replaced by a table with appropriate statistical analysis.
Table 7 has to clearly explain who went into remission after what time, and needs to explain who met the criteria for resistance regarding prolactin levels or tumor size. Table 8 needs to give dose of BC per day and CB per week. Duration does not make sense. Those who were successful would be maintained on drug, those not successful would be sent to surgery.
The definitions need to be reconsidered. Remission would mean not requiring any medication after either medical or surgical treatment. It is not clear how many went into remission and did not require drug therapy, nor how long they were treated before they went into remission. Favorable outcome for tumor size, especially in larger tumors, must be evaluated at six to nine months, not at three months. Relapse would be someone who has been successfully treated with medication or surgery and has been stable and then has a recurrence. Resistance to dopamine agonists should be defined as resistance to achieving normal prolactin levels after three months. The effects of each dopaminergic drug should be evaluated. Resistance to tumor size reduction should be evaluated at six months and also according to the therapy.
Comments on the Quality of English LanguageThe terminology "evolution" is incorrect. The use of the word "objectified" is inappropriate.
Author Response
Response to reviewer 1
Comment: "Table 1 with the corresponding result section is meaningless without a hypothesis on weight. They have acknowledged that G3 is from an older group, and the observation without a hypothesis has no prognostic value. Table 4 also should be eliminated as there is no hypothesis for metabolic syndrome to separate the three groups."
Response: We appreciate your thorough review of our manuscript. However, we respectfully disagree with your assertion regarding the absence of hypotheses and the prognostic value of our observations in Table 1 and Table 4 (Now Table 2).
In the discussion section, we highlighted the observed metabolic differences between groups, supported by statistical significance and corroborated by existing literature:
"On the metabolic level, in our series, obese and/or overweight patients were statistically more frequent in G3 versus G1 and G2. Likewise, waist size means were greater in G3 in both sexes. According to two series in the literature [10,13], patients with a higher prolactin level have a higher BMI and WS. The correlation between these parameters is not significant. However, the decrease in BMI and WS after 2 to 6 months of treatment is significant in men [11,13,14]. In the literature, hyperprolactinemia is frequently associated with a carbohydrate disorder [10,13]. Several physiopathological phenomena are suggested concerning this subject. Indeed, prolactin causes a state of insulin resistance (IR) and hyperinsulinism secondary to the increase in cell mass of Langerhans' islets [29]. Overall, the metabolic syndrome (MS) was more frequently observed in G3 (58.8%) versus G1 (22.2%) and G2 (37.5%) patients with a statistically significant difference. The studies by Auriemma and Dos Santos Silva [13,14] confirm our findings."
This section explicitly addresses both hypotheses underlying Table 1 and Table 4:
- Weight and Waist Size: The correlation between prolactin levels and body mass index (BMI)/waist size (WS) is established in the literature and is a hypothesis we explore in our data.
- Metabolic Syndrome: We observed a higher prevalence of metabolic syndrome in G3, with statistical significance, which aligns with the physiopathological mechanisms described in the literature (e.g., prolactin-induced insulin resistance).
Both findings have implications for clinical practice and contribute to the understanding of hyperprolactinemia's metabolic effects. These observations are neither arbitrary nor prognostically irrelevant, as they reflect documented metabolic phenomena associated with prolactin levels.
We believe that Tables 1 and 2 are essential in illustrating these findings and their statistical significance, which would otherwise be less apparent in the text alone.
We hope this clarifies the purpose and value of these tables and trust that our explanation aligns with the robustness expected in hypothesis-driven research.

Reviewer 2 Report
Comments and Suggestions for Authors
In the manuscript, the authors illustrated studying the epidemiological, clinical and paraclinical characteristics of a Tunisian patients with prolactinoma, looking for correlations between the size of the prolactinoma and the different clinical and paraclinical parameters while analyzing the predictive factors of remission. The results and metrology looked good and attractive. However, there are some issues that authors need to solve.
1. In the manuscript, what were the specific size criteria used to classify tumors into G1, G2, and G3 groups?
2. What were the actual prolactin levels (mean ± SD) for each tumor size group?
3. How did authors differentiate between true dopamine agonist resistance versus poor medication adherence?
4. Were there any correlations between initial prolactin levels and time to normalization within each size group?
5. How did authors get the best-fit lines in the Fig 3-5?
Comments on the Quality of English Languageminor revision
Author Response

(The authors gave the same response as above.)

Round 2
Reviewer 1 Report
Comments and Suggestions for Authors
I cannot find my original review and the authors response. I only find one aspect of the response which is inadequate. A hypothesis and aims are presented in the introduction not the discussion. Please send back my original review, with the corresponding authors' answers so that I may follow the revisions. If the authors want to continue with regard to weight and metabolic syndrome, they need to develop a method for controlling for weight vs age in G3. Are they suggesting that that obesity causes prolactinomas to grow or that prolactinomas are a cause of obesity?
Comments on the Quality of English LanguageI had comments , but they were not addressed
Author Response
Dear Reviewer 1,
Thank you once again for your valuable feedback on our manuscript. We appreciate the time and effort you have dedicated to reviewing our work.
We would like to inform you that all your comments regarding the form and content of the manuscript have been carefully addressed. The revised version of the manuscript is attached for your consideration, along with the present response.
As for the relationship between hyperprolactinemia and insulin resistance, we believe that our data align well with the existing evidence reported by Gierach (2022, DOI: 10.5603/EP.a2022.0075) and Pirchio (2022, PMID: 36237192), among others.
Please do not hesitate to reach out if you have any further questions or require additional clarifications.
Thank you for your constructive insights and support.

Round 3
Reviewer 1 Report
Comments and Suggestions for Authors
The authors have improved the manuscript. Page 5. Figure 1 is duplicated. Page5 figure2. Add to figure legend, ANOVA p value and significant difference among the groups Page 6 delete fig 3. It is confusing. You already have shown differences in Table 3 and correlation analysis is inappropriate. The appropriate test should be ANOVA, which is already done in Table 3. Renumber all tables and figures and check that they match in narrative. It appears that you have eliminated all supplements.
Comments on the Quality of English LanguageEvolution is inappropriate term.
Author Response
Dear Reviewer ,
Thank you once again for your valuable feedback on our manuscript. We appreciate the time and effort you have dedicated to reviewing our work.
We would like to inform you that all your comments regarding the form and content of the manuscript have been carefully addressed. The revised version of the manuscript is attached for your consideration, along with the present response.
Please do not hesitate to reach out if you have any further questions or require additional clarifications.
Thank you for your constructive insights and support.
